# Enforcing Predictive Invariance across Structured Biomedical Domains

## Abstract

Many biochemical applications such as molecular property prediction require models to generalize beyond their training domains (environments). Moreover, natural environments in these tasks are structured, defined by complex descriptors such as molecular scaffolds or protein families. Therefore, most environments are either never seen during training, or contain only a single training example. To address these challenges, we propose a new regret minimization (RGM) algorithm and its extension for structured environments. RGM builds from invariant risk minimization (IRM) by recasting simultaneous optimality condition in terms of predictive regret, finding a representation that enables the predictor to compete against an oracle with hindsight access to held-out environments. The structured extension adaptively highlights variation due to complex environments via specialized domain perturbations. We evaluate our method on multiple applications: molecular property prediction, protein homology and stability prediction and show that RGM significantly outperforms previous state-of-the-art baselines.

## 1 Introduction

In many biomedical applications, training data is necessarily limited or otherwise heterogeneous. It is therefore important to ensure that model predictions derived from such data generalize substantially beyond where the training samples lie. For instance, in molecule property prediction (Wu et al., 2018), models are often evaluated under scaffold split, which introduces structural separation between the chemical spaces of training and test compounds. In protein homology detection (Rao et al., 2019), the split is driven by protein superfamily where entire evolutionary groups are held out from the training set, forcing models to generalize across larger evolutionary gaps.

The key technical challenge is to be able to estimate models that can generalize beyond their training data. The ability to generalize implies a notion of invariance to the differences between the available training data and where predictions are sought. A recently proposed approach known as invariant risk minimization (IRM) (Arjovsky et al., 2019) seeks to find predictors that are simultaneously optimal across different such scenarios (called environments). Indeed, one can apply IRM with environments corresponding to molecules sharing the same scaffold (Bemis & Murcko, 1996) or proteins from the same family (El-Gebali et al., 2019) (see Figure 1). However, this is challenging since, for example, scaffolds are structured objects and can often uniquely identify each example in the training set. It is not helpful to create single-example environments as the model would see any variation from one example to another as scaffold variation.

In this paper, we propose a *regret minimization* algorithm to handle both standard and structured environments. The basic idea is to simulate unseen environments by using part of the training set as held-out environments $E_e$. We quantify generalization in terms of regret — the difference between the losses of two auxiliary predictors trained with and without examples in $E_e$. This imposes a stronger constraint on $\phi$ and avoids some undesired representations admitted by IRM. For the structured environments like molecular scaffolds, we simulate unseen environments by perturbing the representation $\phi$. The perturbation is defined as the gradient of an auxiliary scaffold classifier with respect to $\phi$. The difference between the original and perturbed representation highlights the scaffold variation to the model. Its associated regret measures how well a predictor trained without perturbation generalizes to the perturbed examples. The goal is to characterize the scaffold variation without explicitly creating an environment for every possible scaffold.

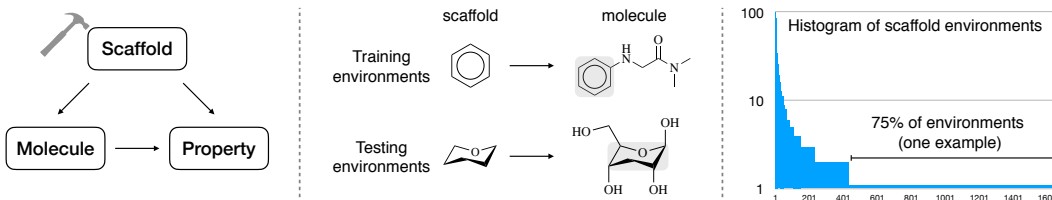

Figure 1: *Left*: Data generation process for molecule property prediction. Training and test environments are generated by controlling the scaffold variable. *Middle*: Scaffold is a subgraph of a molecular graph with its side chains removed. *Right*: In a toxicity prediction task (Wu et al., 2018), there are 1600 scaffold environments with 75% of them having a single example.

Our methods are evaluated on real-world datasets such as molecule property prediction and protein classification. We compare our model against multiple baselines including IRM, MLDG (Li et al., 2018a) and CrossGrad (Shankar et al., 2018). On the QM9 dataset (Ramakrishnan et al., 2014), we outperform the best baseline by a wide margin across multiple properties (41.7 v.s 52.3 average MAE) under an extrapolation evaluation. On a protein stability dataset (Rocklin et al., 2017), we achieve new state-of-the-art results compared to Rao et al. (2019) (0.79 v.s. 0.73 spearman's $\rho$).

## 2 RELATED WORK

**Generalization challenges in biomedical applications** The challenges of generalization have been extensively documented in this area. For instance, Yang et al. (2019); Rao et al. (2019); Hou et al. (2018) have demonstrated that state-of-the-art models exhibit drop in performance when tested under scaffold or protein family split. De facto, the scaffold split and its variants (Feinberg et al., 2018) are used so commonly in cheminformatics as they emulate temporal evaluation adopted in pharmaceutical industry. Therefore, the ability to generalize to new scaffold or protein family environments is the key for practical usage of these models. Moreover, input objects in these domains are typically structured (e.g., molecules are represented by graphs (Duvenaud et al., 2015; Dai et al., 2016; Gilmer et al., 2017)). This characteristic introduces unique challenges with respect to the environment definition for IRM style algorithms.

**Invariance** Prior work has sought generalization by enforcing an appropriate invariance constraint over learned representations. For instance, domain adversarial network (DANN) (Ganin et al., 2016; Zhao et al., 2018) enforces the latent representation $Z = \phi(X)$ to have the same distribution across different environments $E$ (i.e, $Z \perp E$). However, this forces predicted label distribution $P(Y|Z)$ to be the same across all the environments (Zhao et al., 2019). Long et al. (2018); Li et al. (2018c); Combes et al. (2020) extends the invariance criterion by conditioning on the label in order to address the label shift issue of DANN. Invariant risk minimization (IRM) (Arjovsky et al., 2019) seeks a different notion of invariance. Instead of aligning distributions of $Z$, IRM requires that the predictor $f$ operating on $Z = \phi(X)$ is simultaneously optimal across different environments. The associated independence is $Y \perp E \mid Z$. Various work (Krueger et al., 2020; Chang et al., 2020) has sought to extend IRM. We focus on the structured setting, where most of the environments can uniquely specify $X$ in the training set. As a result, $E$ would act similarly to $X$. In the extreme case, the IRM principle reduces to $Y \perp X \mid Z$, which is not the desired invariance criterion. We propose to address this issue by introducing domain perturbation to adaptively highlight the structured variation.

**Domain generalization** These methods seek to learn models that generalize to new domains (Muandet et al., 2013; Ghifary et al., 2015; Motiian et al., 2017; Li et al., 2017; 2018b). Domain generalization methods can be roughly divided into three categories: *domain adversarial training* (Ganin et al., 2016; Tzeng et al., 2017; Long et al., 2018), *meta-learning* (Li et al., 2018a; Balaji et al., 2018; Li et al., 2019a;b; Dou et al., 2019) and *domain augmentation* (Shankar et al., 2018; Volpi et al., 2018). Our method resembles meta-learning based methods in that we create held-out environments to simulate domain shift during training. However, our objective seeks to reduce the regret between predictors trained with or without access to the held-out environments.

Existing domain generalization benchmarks assume that each domain contains sufficient amounts of data. We focus on a different setting where most of the environments contain only few (or single)

examples since they are defined by structured descriptors. This setting often arises in chemical and biological applications (see Figure 1). Similar to data augmentation method in Shankar et al. (2018), our structured RGM also creates perturbed examples based on domain-guided perturbations. However, our method operates over learned representations since our inputs are discrete. Moreover, the perturbed examples are only used to regularize the feature extractor $\phi$ via the regret term.

## 3 REGRET MINIMIZATION

To introduce our method, we start with a standard setting where the training set $\mathcal{D}$ is comprised of $n$ environments $\mathcal{E} = \{E_1, \cdots, E_n\}$ (Arjovsky et al., 2019). Each environment $E_i$ consists of examples $(x, y)$ randomly drawn from some distribution $\mathcal{P}_i$. Assuming that new environments we may encounter at test time exhibit similar variability as the training environments, our goal is to train a model that generalizes to such new environments $E_{\text{test}}$. Suppose our model consists of two components $f \circ \phi$, where the predictor $f$ operates on the feature extractor $\phi$. Let $\mathcal{L}^e(f \circ \phi) = \sum_{(x,y) \in E_e} \ell(y, f(\phi(x)))$ be its empirical loss in environment $E_e$ and $\mathcal{L}(f \circ \phi) = \sum_e \mathcal{L}^e(f \circ \phi)$. IRM learns $\phi$ and $f$ such that $f$ is simultaneously optimal in all training environments:

$$\min_{\phi, f} \mathcal{L}(f \circ \phi) \qquad \text{s.t.} \quad \forall e : f \in \arg\min_h \mathcal{L}^e(h \circ \phi) \tag{1}$$

One possible way to solve this objective is through Lagrangian relaxation:

$$\min_{\phi, f} \mathcal{L}(f \circ \phi) + \sum_e \lambda_e \big( \mathcal{L}^e(f \circ \phi) - \min_h \mathcal{L}^e(h \circ \phi) \big) \tag{2}$$

The regularizer $\mathcal{L}^e(f \circ \phi) - \min_h \mathcal{L}^e(h \circ \phi)$ measures the performance gap between $f$ and the best predictor $\hat{h} \in F_e(\phi) = \arg\min_h \mathcal{L}^e(h \circ \phi)$ specific to environment $E_e$. Note that both $f$ and $\hat{h}$ are trained and evaluated on examples from environment $E_e$. This motivates us to replace the regularizer with a predictive regret. Specifically, for each environment $E_e$, we define the associated regret $\mathcal{R}^e(\phi)$ as the difference between the losses of two auxiliary predictors trained *with* and *without* access to examples $(x, y) \in E_e$:

$$\mathcal{R}^e(\phi) = \mathcal{L}^e(f_{-e} \circ \phi) - \min_{h \in \mathcal{F}} \mathcal{L}^e(h \circ \phi) = \mathcal{L}^e(f_{-e} \circ \phi) - \mathcal{L}^e(f_e \circ \phi) \tag{3}$$

where the two auxiliary predictors are obtained from (assuming $\mathcal{F}$ is bounded and closed):

$$f_e \in F_e(\phi) = \arg\min_{h \in \mathcal{F}} \mathcal{L}^e(h \circ \phi) \qquad f_{-e} \in F_{-e}(\phi) = \arg\min_{h \in \mathcal{F}} \sum_{k \neq e} \mathcal{L}^k(h \circ \phi) \tag{4}$$

The *oracle predictor* $f_e$ is trained on environment $E_e$, while $f_{-e}$ uses the rest of the environments $\mathcal{E} \setminus \{E_e\}$ for training but is tested on $E_e$. Note that $\mathcal{R}^e(\phi)$ does not depend on the predictor $f$ we are seeking to estimate; it is a function of the representation $\phi$ as well as the two auxiliary predictors $f_{-e}$ and $f_e$. For notational simplicity, we have omitted $R^e(\phi)$'s dependence on $f_{-e}$ and $f_e$. Since both predictors are evaluated on the same set of training examples in $E_e$, we immediately have

**Proposition 1.** *The regret $\mathcal{R}^e(\phi)$ is always non-negative for any representation $\phi$.*

The proof is straightforward since $f_e$ is the minimizer of $\mathcal{L}^e(f' \circ \phi)$ and both $f_e$ and $f_{-e}$ are drawn from the same parametric family $\mathcal{F}$. The overall regret $\mathcal{R}(\phi) = \sum_e \mathcal{R}^e(\phi)$ expresses our stated goal of finding a representation $\phi$ that generalizes to each held-out environment. Our regret minimization (RGM) objective regularizes the empirical loss with a regret term weighted by $\lambda$:

$$\mathcal{L}_{\text{RGM}} = \mathcal{L}(f \circ \phi) + \lambda \sum_e \mathcal{R}^e(\phi) \tag{5}$$

### 3.1 COMPARISON WITH IRM

Compared to IRM, the proposed RGM objective imposes a stronger constraint on $\phi$ since $f_{-e}$ is not trained on $E_e$. To show this formally, let $F_e(\phi), F_{-e}(\phi)$ be the set of optimal predictors in $E_e$ and $\mathcal{E} \setminus \{E_e\}$ respectively as defined in Eq.(4). Since $\mathcal{R}^e(\phi) = 0 \Leftrightarrow f_{-e} \in F_e(\phi)$ and $f_{-e}$ is chosen arbitrarily from $F_{-e}(\phi)$, the constrained form of the RGM objective can be stated as

$$\min_{\phi, f} \mathcal{L}(f \circ \phi) \qquad \text{s.t.} \quad \forall e : F_{-e}(\phi) \subseteq F_e(\phi) \tag{6}$$

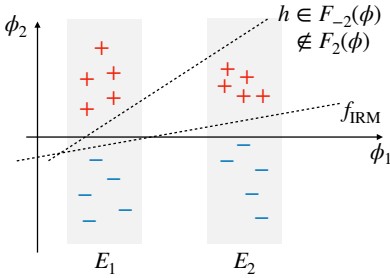

Figure 2: A counterexample illustrating that $\Phi_{\text{IRM}} \not\subseteq \Phi_{\text{RGM}}$. The environments are generated by different translations of $X_1$. For the identity mapping $\phi(X) = (X_1, X_2)$ and the true hypothesis is $\mathbb{I}[X_2 > 0]$. There exists a predictor $f_{\text{IRM}}$ which is simultaneously optimal in all environments. In contrast, $\phi$ is not feasible under RGM because there is a linear classifier $h \in F_{-2}(\phi)$ that is optimal in environment $E_1$ but performs poorly in environment $E_2$.

The analogous IRM constraints are $f \in \cap_e F_e(\phi)$ and $\cap_e F_e(\phi) \neq \emptyset$. Suppose both IRM and RGM constraints are feasible and let $\mathcal{L}^*_{\text{IRM}}, \mathcal{L}^*_{\text{RGM}}$ be their optimal loss respectively. Consider the set of optimal features under both objectives:

$$\Phi_{\text{IRM}} = \{\phi \mid \min_{f \in \cap_e F_e(\phi)} \mathcal{L}(f \circ \phi) = \mathcal{L}^*_{\text{IRM}}, \cap_e F_e(\phi) \neq \emptyset\} \tag{7}$$

$$\Phi_{\text{RGM}} = \{\phi \mid \min_{f \in \mathcal{F}} \mathcal{L}(f \circ \phi) = \mathcal{L}^*_{\text{RGM}}, \forall e : F_{-e}(\phi) \subseteq F_e(\phi)\} \tag{8}$$

**Proposition 2.** *Assuming two environments, if $\mathcal{L}^*_{\text{RGM}} = \mathcal{L}^*_{\text{IRM}}$, then $\Phi_{\text{RGM}} \subseteq \Phi_{\text{IRM}}$. The converse $\Phi_{\text{IRM}} \subseteq \Phi_{\text{RGM}}$ does not hold in general.*

While limited to two environments, the proposition suggests that RGM imposes stronger constraints on $\phi$. Figure 2 shows a counterexample illustrating that $\Phi_{\text{IRM}} \not\subseteq \Phi_{\text{RGM}}$. Suppose there are two environments generated by translation of $X_1$ and the true hypothesis is $\mathbb{I}[X_2 > 0]$. The identity mapping $\phi(X) = (X_1, X_2)$ is not translation invariant, but $\phi \in \Phi_{\text{IRM}}$ because there exists a predictor $f_{\text{IRM}}$ that is simultaneously optimal in all environments. On the other hand, $\phi$ is not feasible under RGM because there is a linear classifier $h \in F_{-2}(\phi)$ that is optimal in $E_1$ but suboptimal in $E_2$, violating the RGM constraint $F_{-2}(\phi) \subseteq F_2(\phi)$. Thus $\phi \notin \Phi_{\text{RGM}}$.

To see why it would be helpful to add a stronger constraint on $\phi$, consider the following data generation process where the environment $e$ can be inferred from input $x$ alone:

$$p(x, y, e) = p(e)p(x|e)p(y|x, e); \qquad p(y|x, e) = p(y|x, e(x)) \tag{9}$$

For molecules and proteins, this assumption is often valid because the environment labels (scaffolds, protein families) typically depend on $x$ only. We call $\phi$ *label-preserving* if it retains all the information about the label: $p(y|\phi(x)) = p(y|x, e)$. Such representation may not generalize to new environments given the dependence on $e$ through $\phi$. However, we can show that for any label-preserving $\phi$, its associated ERM optimal predictor also satisfies the IRM constraints:

**Proposition 3.** *For any label-preserving $\phi$ with $p(y|\phi(x)) = p(y|x, e)$, its associated ERM optimal predictor $f^*$ satisfies the IRM constraint. Moreover, if $\phi \in \Phi_{\text{IRM}}$, $f^* \circ \phi$ is optimal under IRM.*

While IRM constraints are vacuous for any label-preserving $\phi$, this is not necessarily the case with RGM constraints. Consider, for example, the counterexample in Figure 4. The identity mapping $\phi(X) = (X_1, X_2)$ is label-preserving since it retains all the input information. However, $\phi$ is infeasible under RGM.

### 3.2 STRUCTURED ENVIRONMENTS

Now let us consider a more challenging setting, where the environments $\{E_k\}$ are structured (i.e., $k$ is a structured object rather than an integer). Formally, the training set comes in the form $\mathcal{D} = \{(x_i, y_i, s_i)\}$, where $s_i$ is the structured environment label of $(x_i, y_i) \in E_{s_i}$. For instance, in molecule property prediction, $s_i$ is defined as the Murcko scaffold (i.e., subgraph) of molecule $x_i$. It is hard to model scaffolds as standard environments because they are structured descriptors and often uniquely identify each molecule in the training set (Figure 1). When an environment has only one molecule, the model cannot decide which subgraph of that molecule is the right scaffold. Thus, creating single-example environments is not helpful for domain generalization.

Alternatively, we can describe scaffold variation by perturbation in the representation $\phi$. The idea is to create a perturbed instance $\tilde{x}_i$ for each example $(x_i, y_i, s_i)$ so that the difference between $x_i$ and $\tilde{x}_i$ highlights how scaffold information has changed in the representation. Specifically, the perturbation

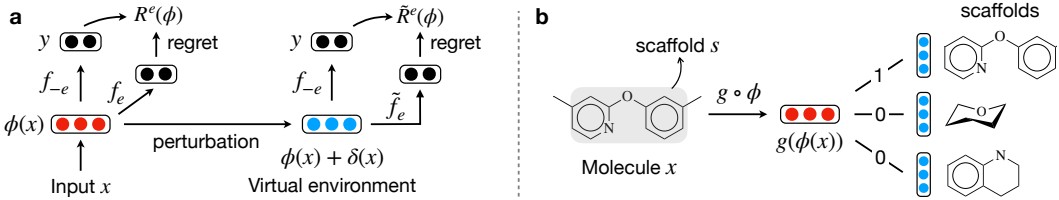

Figure 3: a) Structured RGM: we introduce additional oracle predictors $\tilde{f}_e$ for the perturbed inputs; b) In molecule tasks, the scaffold classifier $g$ is trained by negative sampling.

---

**Algorithm 1** Structured RGM: Forward Pass

---

1: **for** each environment $E_e \in \mathcal{E}$ **do**
2:     Sample a minibatch $B_e$ from environment $E_e$
3:     Compute scaffold classification loss $\mathcal{L}_g(g \circ \phi)$ over $B_e$.
4:     Construct perturbed examples $\tilde{B}_e$ from $B_e$ via gradient perturbation (see Eq.(10)).
5:     Compute empirical loss $\mathcal{L}(f \circ \phi)$ on $B_e$.
6:     Compute auxiliary predictor loss $\mathcal{L}^{-e}(f_{-e} \circ \phi)$ on $B_{-e}$.
7:     Compute oracle predictor losses $\mathcal{L}^e(f_e \circ \phi)$ and $\mathcal{L}^e(\tilde{f}_e \circ (\phi + \delta))$ on $B_e$ and $\tilde{B}_e$.
8:     Compute regret terms $\mathcal{R}^e(\phi), \mathcal{R}^e(\phi + \delta)$ on $B_e$ and $\tilde{B}_e$.
9: **end for**

---

$\delta(x_i)$ is defined through a parametric scaffold classifier $g$ built on top of the representation $\phi$.[1] The associated scaffold classification loss is $\ell(s_i, g(\phi(x_i)))$. Given that our inputs are discrete, we define the perturbation $\delta$ as the gradient with respect to the continuous representation $\phi$:

$$\phi(\tilde{x}_i) := \phi(x_i) + \delta(x_i) = \phi(x_i) + \alpha \nabla_z \ell(s_i, g(z))|_{z=\phi(x_i)} \qquad (10)$$

where $\alpha$ is a step size parameter. The perturbation is specifically designed to contain less information about the scaffold $s_i$, and we require that the model should not be affected by this variation in the representation. Since these perturbations introduce additional simulated test scenarios that we wish to generalize to, we propose to regularize our model also based on regret associated with perturbed inputs. Similar to Eq.(3), the regret corresponding to perturbed inputs is defined as $\mathcal{R}^e(\phi + \delta)$:

$$\mathcal{R}^e(\phi + \delta) = \mathcal{L}^e(f_{-e} \circ (\phi + \delta)) - \min_h \mathcal{L}^e(h \circ (\phi + \delta)) \qquad (11)$$

$$\mathcal{L}^e(h \circ (\phi + \delta)) = \sum_{(x_i, y_i) \in E_e} \ell\big(y_i, h(\phi(x_i) + \delta(x_i))\big) \qquad (12)$$

This introduces a new oracle predictor $\tilde{f}_e = \arg\min_h \mathcal{L}^e(h \circ (\phi + \delta))$ for each environment $E_e$ (see Figure 3a). Note that $f_{-e}$ is the same auxiliary predictor as before. It minimizes a separate objective $\mathcal{L}^{-e}(f_{-e} \circ \phi)$, which does *not* include the perturbed examples.

The structured RGM (SRGM) objective $\mathcal{L}_{\text{SRGM}}$ augments the basic RGM with additional regret terms as well as the scaffold classification loss $\mathcal{L}_g(g \circ \phi)$:

$$\mathcal{L}_{\text{SRGM}} = \mathcal{L}(f \circ \phi) + \lambda_g \mathcal{L}_g(g \circ \phi) + \lambda \sum_e \sum_{\psi \in \{0, \delta\}} \mathcal{R}^e(\phi + \psi) \qquad (13)$$

$$\mathcal{L}_g(g \circ \phi) = \sum_{(x_i, y_i, s_i) \in \mathcal{D}} \ell\big(s_i, g(\phi(x_i))\big) \qquad (14)$$

The forward pass of SRGM is shown in Algorithm 1. Since $s$ is a structured object with a large number of possible values, we train the classifier $g$ with negative sampling (Figure 3b). Note that $\phi$ is also updated to partially optimize $\mathcal{L}_g$. This is necessary to ensure that the scaffold classifier operating on $\phi$ has enough information to introduce a reasonable gradient perturbation $\delta(x)$. This trade-off keeps some scaffold information in $\phi$ while ensuring, via the associated regret terms, that this information is not strongly relied upon. The effect of this design choice is studied in our experiments.

---

[1]Our method is introduced using scaffolds as examples. It can be applied to other structured environments like protein families by simply replacing the scaffold classifier with a protein family classifier.

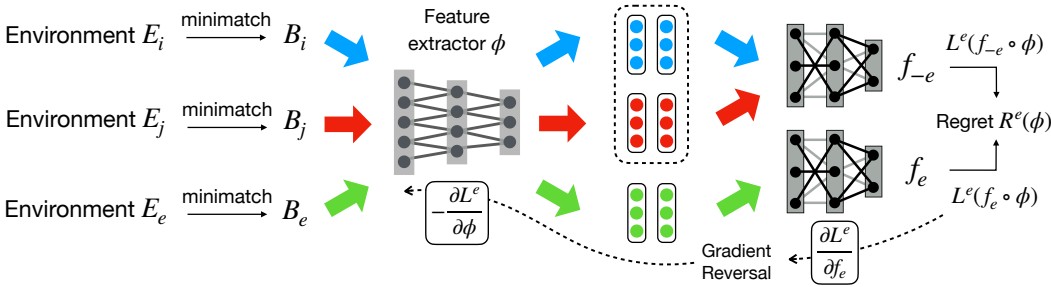

Figure 4: In the RGM forward pass, we sample a minibatch $B_e$ from each environment $E_e$ and compute regret $R^e(\phi)$. In the backward pass, the gradient of $\mathcal{L}^e(f_e \circ \phi)$ goes through a gradient reversal layer (Ganin et al., 2016) which negates the gradient during back-propagation.

### 3.3 OPTIMIZATION

The standard RGM objective in Eq.(5) can be viewed as finding a stationary point of a multi-player game between $f$, $\phi$ as well as the auxiliary predictors $\{f_{-e}\}$ and $\{f_e\}$. Our predictor $f$ and representation $\phi$ find their best response strategies by minimizing

$$\min_{f,\phi} \left\{ \mathcal{L}(f \circ \phi) + \lambda \sum_e \left( \mathcal{L}^e(f_{-e} \circ \phi) - \mathcal{L}^e(f_e \circ \phi) \right) \right\} \tag{15}$$

while the auxiliary predictors minimize

$$\min_{f_{-e}} \mathcal{L}^{-e}(f_{-e} \circ \phi) \quad \text{and} \quad \min_{f_e} \mathcal{L}^e(f_e \circ \phi) \quad \forall e \tag{16}$$

This multi-player game can be optimized by stochastic gradient descent. Since $f_e$ and $\phi$ optimizes $\mathcal{L}^e(f_e \circ \phi)$ in opposite directions, we introduce a gradient reversal layer (Ganin et al., 2016) between $\phi$ and $f_e$. This allows us to update all the players in a single forward-backward pass (see Figure 4). In each step, we simultaneously update all the players with learning rate $\eta$:

$$f \leftarrow f - \eta \nabla_f \mathcal{L}(f \circ \phi) \qquad\qquad \phi \leftarrow \phi - \eta \nabla_\phi \mathcal{L}(f \circ \phi) - \eta \lambda \sum_e \nabla_\phi \mathcal{R}^e(\phi)$$

$$f_{-e} \leftarrow f_{-e} - \eta \nabla \mathcal{L}^{-e}(f_{-e} \circ \phi) \qquad f_e \leftarrow f_e - \eta \nabla \mathcal{L}^e(f_e \circ \phi) \quad \forall e$$

where $\mathcal{L}^{-e}(f_{-e} \circ \phi) = \sum_{k \neq e} \mathcal{L}^k(f_{-e} \circ \phi)$. In each step, we sample minibatches $B_1, \cdots, B_n$ from each environment $E_1, \cdots, E_n$. The loss $\mathcal{L}(f \circ \phi)$ is computed over all the minibatches $\bigcup_k B_k$, while $\mathcal{L}^{-e}(f_{-e} \circ \phi)$ is computed over minibatches $B_{-e} = \bigcup_{k \neq e} B_k$. The regret term $R^e(\phi)$ is evaluated based on examples in $B_e$ only.

For structured RGM, its optimization rule is analogous to RGM, with additional gradient updates for the oracle predictors $\tilde{f}_e$ and scaffold classifier $g$ (see Appendix A.4). While the perturbation $\delta$ is defined on the basis of $\phi$ and $g$, we do not include the dependence during back-propagation as incorporating this higher order gradient does not improve our empirical results.

## 4 EXPERIMENTS

Our methods (RGM and SRGM) are evaluated on real-world applications such as molecular property prediction, protein homology and stability prediction. Our baselines include:

- Standard empirical risk minimization (ERM) trained on aggregated environments;
- Domain adversarial training methods including DANN (Ganin et al., 2016) and CDAN (Long et al., 2018), which seek to learn domain-invariant features;
- IRM (Arjovsky et al., 2019) requiring the model to be simultaneously optimal in all environments;
- MLDG (Li et al., 2018a), a meta-learning method which simulates domain shift by dividing training environments into meta-training and meta-testing;
- CrossGrad (Shankar et al., 2018) which augments the training set with domain-guided perturbations of inputs. Since our inputs are discrete, we perform perturbation on the representation instead.

Table 1: Mean absolute error (MAE) on the QM9 dataset. Models are evaluated under scaffold split. Due to space limit, we only show standard deviation for the top three methods in subscripts.

| Property | Categorical methods | | | | | | Structured methods | |
|---|---|---|---|---|---|---|---|---|
| | ERM | DANN | CDAN | IRM | MLDG | RGM | CrossGrad | SRGM |
| mu | 0.736 | 0.709 | 0.748 | 1.059 | 0.745 | **0.682**$_{(.057)}$ | 0.745$_{(.077)}$ | 0.720$_{(.089)}$ |
| alpha | 3.455 | 3.525 | 3.668 | 3.711 | 3.261 | **2.600**$_{(.016)}$ | 3.563$_{(1.44)}$ | 2.694$_{(.018)}$ |
| HOMO | 0.011 | 0.011 | 0.012 | 0.011 | **0.010** | 0.011$_{(.002)}$ | 0.012$_{(.002)}$ | 0.011$_{(.002)}$ |
| LUMO | 0.020 | 0.020 | 0.021 | 0.021 | 0.020 | 0.020$_{(.002)}$ | **0.017**$_{(.002)}$ | 0.019$_{(.002)}$ |
| gap | 0.021 | 0.020 | 0.021 | 0.022 | 0.021 | 0.020$_{(.002)}$ | **0.019**$_{(.002)}$ | **0.019**$_{(.001)}$ |
| R2 | 119.5 | 117.1 | 120.4 | 174.2 | 110.8 | 113.9$_{(6.10)}$ | 112.2$_{(12.3)}$ | **100.5**$_{(7.53)}$ |
| ZPVE | 0.008 | 0.008 | 0.009 | 0.009 | 0.009 | 0.008$_{(.002)}$ | 0.008$_{(.001)}$ | **0.007**$_{(.001)}$ |
| Cv | 1.917 | 1.960 | 2.093 | 2.344 | 2.029 | 2.268$_{(.417)}$ | **1.702**$_{(.330)}$ | 2.133$_{(.423)}$ |
| U0 | 17.50 | 18.50 | 18.25 | 16.21 | 16.24 | 14.93$_{(2.96)}$ | 20.11$_{(5.08)}$ | **13.97**$_{(1.32)}$ |
| U | 20.11 | 20.51 | 20.41 | 16.72 | 17.65 | 14.39$_{(2.57)}$ | 14.52$_{(1.31)}$ | **12.67**$_{(0.82)}$ |
| H | 17.40 | 17.34 | 18.11 | 16.53 | 14.77 | 13.97$_{(1.01)}$ | 18.55$_{(3.59)}$ | **12.80**$_{(1.21)}$ |
| G | 17.67 | 18.63 | 19.09 | 17.68 | 16.14 | 13.53$_{(1.27)}$ | 17.95$_{(5.12)}$ | **13.15**$_{(1.18)}$ |

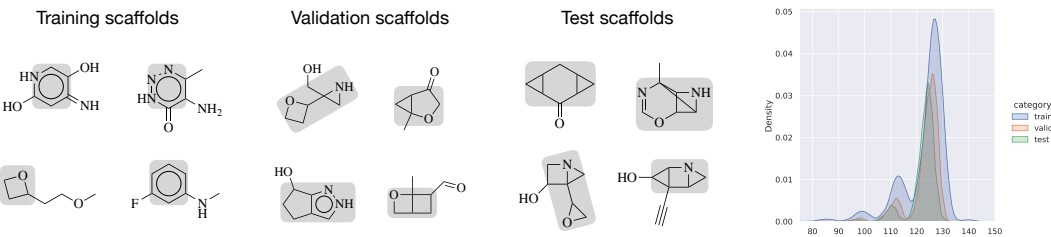

Figure 5: Examples of scaffolds in the QM9 dataset (highlighted in grey). We split the data based on scaffold complexity. Thus, the test scaffolds are structurally distinct from scaffolds in the training set. As shown in the right figure, the molecular weight distribution of training, validation and test sets are similar. This shows that scaffold complexity split is more realistic than molecular weight split.

These methods fall into two categories. SRGM and CrossGrad are *structured* methods as they can leverage the structural information of the environment (e.g., scaffold). RGM and other methods are *categorical* methods since they do not utilize the structure and simply treat each environment as a set.

### 4.1 MOLECULAR PROPERTY PREDICTION

**Data** The training data consists of $\{(x_i, y_i, s_i)\}$, where $x_i$ is a molecular graph, $y_i$ is its property and $s_i$ is its scaffold. We adopt four datasets from the MoleculeNet benchmark (Wu et al., 2018):

- QM9 is a regression dataset of 134K organic molecules with up to 9 heavy atoms. Each molecule is labeled with 12 quantum mechanical properties.
- HIV is a classification dataset of 42K molecules. Each molecule is associated with a binary label indicating whether it is an HIV inhibitor.
- Tox21 is a classification dataset of 8.8K molecules. Each compound has 12 binary labels for toxicity measurements.
- The blood-brain barrier penetration (BBBP) dataset contains 2K molecules. Each molecule is labeled with a binary permeability label.

**Data split** To test whether a model generalizes to new domains, it is important to create a test set that is distributionally distinct from the training set. Scaffold split (Wu et al., 2018) is a common framework for this purpose. Molecules are clustered based on its Bemis-Murcko scaffold (Bemis & Murcko, 1996) and a random subset of scaffolds are selected into a test set. However, this approach degenerates to random split when most scaffold clusters contain only one molecule (see Figure 1). To address this issue, Feinberg et al. (2019) proposed molecular weight split, where test molecules

Table 2: *Left*: Results on molecule and protein datasets. CrossGrad and SRGM use the structure of environments (scaffolds or protein superfamily) while others do not. *Right*: Ablation study of SRGM. Detach=✓ means we do not update $\phi$ to optimize the scaffold (or protein superfamily) classification loss $\mathcal{L}_g$. $\text{Acc}_S$ stands for the scaffold/protein superfamily classification accuracy. Property is the property prediction performance (AUROC for molecules, top-1 accuracy for protein).

| | HIV | Tox21 | BBBP | Protein | | detach | $\text{Acc}_S$ | property |
|---|---|---|---|---|---|---|---|---|
| ERM | $0.715_{(.032)}$ | $0.641_{(.004)}$ | $0.854_{(.024)}$ | 20.9% | HIV | ✓ | 88.1% | 0.736 |
| DANN | $0.727_{(.029)}$ | $0.639_{(.006)}$ | $0.857_{(.016)}$ | 22.3% | | ✗ | **99.4%** | **0.751** |
| CDAN | $0.735_{(.013)}$ | $0.639_{(.008)}$ | $0.853_{(.022)}$ | 21.9% | Tox21 | ✓ | 92.2% | 0.640 |
| IRM | $0.747_{(.007)}$ | $0.632_{(.011)}$ | $0.862_{(.030)}$ | 21.0% | | ✗ | **97.1%** | **0.649** |
| MLDG | $0.724_{(.036)}$ | $0.637_{(.007)}$ | $0.849_{(.016)}$ | 22.0% | BBBP | ✓ | 73.7% | 0.871 |
| RGM | **$0.751_{(.029)}$** | $0.637_{(.010)}$ | $0.858_{(.021)}$ | 23.4% | | ✗ | **94.0%** | **0.891** |
| CrossGrad | $0.746_{(.015)}$ | $0.644_{(.005)}$ | $0.884_{(.030)}$ | 20.9% | Protein | ✓ | 29.3% | 21.9% |
| SRGM | **$0.751_{(.014)}$** | **$0.649_{(.009)}$** | **$0.891_{(.025)}$** | **23.8%** | | ✗ | **33.5%** | **23.8%** |

are much bigger than molecules in the training set. While this creates strong structural distinction between the training and test sets, it is not as realistic as the scaffold split.

Given these observations, we propose a variant of scaffold split called *scaffold complexity split*. We define the complexity of a scaffold as the number of cycles in the scaffold graph. Specifically, we put a scaffold in the test set if its scaffold complexity is greater than $\tau$ and the training set if it is less than $\tau$. We set $\tau = 2$ for QM9 and $\tau = 4$ for other datasets. As shown in Figure 5, this forces the test scaffolds to be structurally different from the training scaffolds. It is also more realistic than the molecular weight split since the molecular weight distribution of training and test sets are similar.

**Model** The molecule encoder $\phi$ is a graph convolutional network (Yang et al., 2019) which translates a molecular graph into a continuous vector. The predictor $f$ is a two-layer MLP that takes $\phi(x)$ as input and predicts the label. The scaffold classifier $g$ is also a two-layer MLP trained by negative sampling since scaffold is a combinatorial object with a large number of possible values. Specifically, for a given molecule $x_i$ with scaffold $s_i$, we randomly sample $K$ other molecules and take their associated scaffolds $\{s_k\}$ as negative classes. Details of model architecture and hyper-parameters are discussed in the appendix.

**Results** Following Wu et al. (2018), we report mean absolute error (MAE) for QM9 and AUROC for other datasets. All the results are averaged across five independent runs. Our results on the QM9 dataset are shown in Table 1. RGM outperforms other categorical methods and demonstrates clear improvement on six properties (mu, alpha, U0, U, H, G). SRGM outperforms all baselines on seven properties, with a significant error reduction on R2, U0, U, H and G (3-10%). Compared to RGM, SRGM performs better on all properties except mu and alpha. On the three classification datasets, SRGM also achieves state-of-the-art compared to all the baselines (see Table 2). These results confirm the advantage of exploiting the structure of environments.

## 4.2 PROTEIN HOMOLOGY PREDICTION

**Data** The protein homology dataset (Fox et al., 2013; Rao et al., 2019) consists of tuples $\{(x_i, y_i, s_i)\}$, where $x_i$ is a protein represented as sequence of amino acids, $y_i$ its fold label and $s_i$ its superfamily label. The task is to predict the fold label $y_i$. There are 1195 fold classes and 1823 protein superfamilies in total. Around 1200 superfamilies have less than 10 instances in the training set.

**Data split** Provided by Rao et al. (2019), the dataset consists of 12K instances for training, 736 for validation and 718 for testing. The dataset is split based on protein superfamilies. As a result, proteins in the test set are structurally distinct from the training set, requiring models to generalize across large evolutionary gaps.

**Model** Our protein encoder $\phi$ is a pre-trained BERT model (Rao et al., 2019). To generate a sequence-length invariant protein embedding, we simply take the mean of all the vectors output by BERT. The predictor $f$ is a linear function that takes $\phi(x)$ as input and predicts its fold class. The superfamily classifier $g$ is a two-layer MLP. The hyperparameters are listed in the appendix.

Table 3: Comparison between SRGM with different perturbations on the QM9 dataset. "Scaffold" means perturbation via the gradient of the scaffold classifier. "Random" means random perturbation.

| SRGM | mu | alpha | homo | lumo | gap | R2 | zpve | Cv | U0 | U | H | G |
|---|---|---|---|---|---|---|---|---|---|---|---|---|
| Scaffold | **0.72** | **2.69** | 0.011 | 0.019 | **0.019** | **100.5** | **0.007** | **2.13** | **13.97** | **12.67** | **12.80** | **13.15** |
| Random | 0.77 | 2.90 | **0.010** | **0.017** | 0.020 | 115.9 | 0.008 | 2.46 | 18.57 | 13.80 | 16.75 | 18.55 |

Table 4: SRGM performance on the QM9 dataset with different number of graph convolutional layers in $\phi$. Adding more layers increases model complexity.

| $\phi$ | mu | alpha | homo | lumo | gap | R2 | zpve | Cv | U0 | U | H | G |
|---|---|---|---|---|---|---|---|---|---|---|---|---|
| 2 layer | **0.69** | 3.06 | **0.010** | **0.014** | **0.018** | 106.1 | 0.014 | 3.33 | 18.72 | 17.72 | 17.43 | 16.67 |
| 3 layer | 0.72 | **2.69** | 0.011 | 0.019 | 0.019 | **100.5** | **0.007** | 2.13 | **13.97** | **12.67** | **12.80** | **13.15** |
| 4 layer | 0.83 | 3.15 | 0.014 | 0.016 | 0.019 | 111.3 | 0.012 | **2.02** | 21.54 | 30.23 | 17.89 | 21.45 |

**Results** Following Rao et al. (2019), we report the top-1 accuracy for homology prediction. Our ERM baseline matches their transformer model performance. As shown in Table 2, both RGM and SRGM outperforms all the baselines (23.8% v.s. 22.3%). The difference between RGM and SRGM is relatively small due to inaccurate superfamily classifier. The top-1 and top-10 superfamily classification accuracy is around 33.5% and 51.0%. Nevertheless, SRGM can still give performance improvement because the gradient perturbation is computed based on the ground truth superfamily label during training. This teacher forcing step helps SRGM to be robust to superfamily variability despite the inaccurate superfamily classifier.

### 4.3 ABLATION STUDY OF SRGM

**Updating $\phi$ for $\mathcal{L}_g$** In section 3.2, we mentioned that the feature extractor $\phi$ is updated to optimize the scaffold (or superfamily) classification loss $\mathcal{L}_g$. To study the effect of this design choice, we evaluate a variant of SRGM called SRGM-detach, where $\phi$ is not updated to optimize the scaffold classification loss. As shown in Table 2 (right), the performance of SRGM-detach is worse than SRGM across the four datasets. This is because the scaffold classifier performs better in SRGM and the gradient $\delta(x)$ reflects the change of scaffold information more accurately.

**Random perturbation** In Table 3, we report the performance of SRGM under random perturbation on the QM9 dataset. Random perturbation performs significantly worse for most of the properties. This shows the importance of the scaffold classifier in SRGM.

**Model complexity** To study how the model complexity of $\phi$ affects the performance of SRGM, we train SRGM under different number of graph convolutional layers on the QM9 dataset. As shown in Table 4, SRGM performs the best when there are three graph convolutional layers, which is adopted in all experiments. In short, SRGM underfits the data when the model is too simple (layer=2) and overfits when the model is too complex (layer=4).

## 5 CONCLUSION

In this paper, we propose regret minimization for generalization across structured biomedical domains such as molecular scaffolds or protein families. We seek to find a representation that enables the predictor to compete against an oracle with hindsight access to unseen domains. Our method significantly outperforms all baselines on real-world biomedical tasks.

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

## A Technical Details

### A.1 Proof of Proposition 1

Note that $\mathcal{L}^e(f \circ \phi)$ is defined on a set of fixed examples in $E_e$. Since $f_e \in \arg\min_{f' \in \mathcal{F}} \mathcal{L}^e(f' \circ \phi)$ and $f_e, f_{-e}$ are in the same parametric family $\mathcal{F}$, we have $\mathcal{R}^e(\phi) = \mathcal{L}^e(f_{-e} \circ \phi) - \mathcal{L}^e(f_e \circ \phi) \geq 0$.

### A.2 Proof of Proposition 2

*Proof.* Consider any representation $\phi^* \in \Phi_{\mathrm{RGM}}$. When there are only two environments $\{E_1, E_2\}$, we have $F_{-2}(\phi^*) = F_1(\phi^*)$ and $F_{-1}(\phi^*) = F_2(\phi^*)$ by definition. Thus the RGM constraint implies

$$F_2(\phi^*) = F_{-1}(\phi^*) \subseteq F_1(\phi^*) \qquad F_1(\phi^*) = F_{-2}(\phi^*) \subseteq F_2(\phi^*)$$

Therefore $F_1(\phi^*) = F_2(\phi^*)$. Since the loss function is non-negative and $\mathcal{F}$ is bounded and closed, $F_1(\phi^*) \neq \emptyset$. Thus, $\cap_e F_e(\phi^*) = F_1(\phi^*) \neq \emptyset$. Now consider any $f \in \cap_e F_e(\phi^*)$. By definition,

$$\forall e : \mathcal{L}^e(f \circ \phi^*) \leq \min_{h \in \mathcal{F}} \mathcal{L}^e(h \circ \phi^*)$$

By summing the above inequality over all environments, we have

$$\sum_e \mathcal{L}^e(f \circ \phi^*) \leq \sum_e \min_{h \in \mathcal{F}} \mathcal{L}^e(h \circ \phi^*) \leq \min_{h \in \mathcal{F}} \sum_e \mathcal{L}^e(h \circ \phi^*)$$

Since $\sum_e \mathcal{L}^e(f \circ \phi^*) = \mathcal{L}(f \circ \phi^*)$, the above inequality implies

$$\mathcal{L}(f \circ \phi^*) \leq \min_{h \in \mathcal{F}} \mathcal{L}(h \circ \phi^*) = \mathcal{L}^*_{\mathrm{RGM}} = \mathcal{L}^*_{\mathrm{IRM}}$$

Thus, $f \circ \phi^*$ is an optimal solution under IRM and $\phi^* \in \Phi_{\mathrm{IRM}}$. $\square$

### A.3 Proof of Proposition 3

*Proof.* Let us recall our assumption of the data generation process:

$$p(x, y, e) = p(e)p(x|e)p(y|x, e); \qquad p(y|x, e) = p(y|x, e(x))$$

Under this assumption, we can rephrase the IRM objective as

$$\min_{f, \phi} \quad \mathbb{E}_e \mathbb{E}_{x|e} \mathbb{E}_{y|x, e} \ell(y, f(\phi(x))) \tag{17}$$

$$\text{s.t.} \quad \mathbb{E}_{x|e} \mathbb{E}_{y|x, e} \ell(y, f(\phi(x))) \leq \min_{f_e} \mathbb{E}_{x|e} \mathbb{E}_{y|x, e} \ell(y, f_e(\phi(x))) \quad \forall e \tag{18}$$

Given any label-preserving representation $\phi(x)$, its ERM optimal predictor is

$$f^*(\phi(x)) = \arg\min_f \mathbb{E}_{y|\phi(x)} \ell(y, f(\phi(x))) \tag{19}$$

To see that $f^*$ is ERM optimal, consider

$$\min_f \mathbb{E}_e \mathbb{E}_{x|e} \mathbb{E}_{y|x, e} \ell(y, f(\phi(x))) \geq \mathbb{E}_e \mathbb{E}_{x|e} \min_f \mathbb{E}_{y|x, e} \ell(y, f(\phi(x))) \tag{20}$$

$$= \mathbb{E}_e \mathbb{E}_{x|e} \min_f \mathbb{E}_{y|\phi(x)} \ell(y, f(\phi(x))) \tag{21}$$

$$= \mathbb{E}_e \mathbb{E}_{x|e} \mathbb{E}_{y|\phi(x)} \ell(y, f^*(\phi(x))) \tag{22}$$

where Eq.(21) holds because $\phi(x)$ is label-preserving. Note that $f^*$ satisfies the IRM constraint because it is simultaneously optimal across all environments:

$$\forall e : \min_{f_e} \mathbb{E}_{x|e} \mathbb{E}_{y|x, e} \ell(y, f_e(\phi(x))) \geq \mathbb{E}_{x|e} \min_{f_e} \mathbb{E}_{y|x, e} \ell(y, f_e(\phi(x))) \tag{23}$$

$$= \mathbb{E}_{x|e} \min_f \mathbb{E}_{y|\phi(x)} \ell(y, f(\phi(x))) \tag{24}$$

$$= \mathbb{E}_{x|e} \mathbb{E}_{y|\phi(x)} \ell(y, f^*(\phi(x))) \tag{25}$$

Moreover, if $\phi \in \Phi_{\mathrm{IRM}}$ is an optimal representation, $f^* \circ \phi$ is an optimal solution of IRM. $\square$

Table 5: Dataset statistics

|            | QM9 | HIV  | Tox21 | BBBP | Homology |
|------------|-----|------|-------|------|----------|
| Training   | 67K | 27K  | 7.6K  | 1275 | 12.3K    |
| Validation | 36K | 7.7K | 776   | 519  | 736      |
| Testing    | 30K | 6.3K | 483   | 248  | 718      |

## A.4 STRUCTURED RGM UPDATE RULE

Since $\tilde{f}_e$ and $\phi$ optimizes $\mathcal{L}(\tilde{f}_e \circ \phi, \tilde{E}_e)$ in different directions, we also introduce a gradient reversal layer between $\phi$ and $\tilde{f}_e$. The SRGM update rule is the following:

$$\phi \leftarrow \phi - \eta \nabla_\phi \mathcal{L}(f \circ \phi) - \eta \lambda_g \nabla_\phi \mathcal{L}_g(g \circ \phi) - \eta \lambda \sum_e \sum_{\psi \in \{0,\delta\}} \nabla_\phi \mathcal{R}^e(\phi + \psi)$$

$$f \leftarrow f - \eta \nabla_f \mathcal{L}(f \circ \phi) \qquad g \leftarrow g - \eta \nabla_g \mathcal{L}_g(g \circ \phi)$$

$$f_e \leftarrow f_e - \eta \nabla \mathcal{L}^e(f_e \circ \phi) \qquad \tilde{f}_e \leftarrow \tilde{f}_e - \eta \nabla \mathcal{L}(\tilde{f}_e \circ (\phi + \delta)) \quad \forall e$$

$$f_{-e} \leftarrow f_{-e} - \eta \nabla \mathcal{L}^{-e}(f_{-e} \circ \phi) \quad \forall e$$

## B EXPERIMENTAL DETAILS

### B.1 MOLECULAR PROPERTY PREDICTION

**Data** The four property prediction datasets are provided in the supplementary material, along with the training/validation/test splits. The size of each training environment, validation and test set are listed in Table 5. The QM9, HIV, Tox21 and BBBP dataset are downloaded from Wu et al. (2018).

**Model Hyperparameters** For the feature extractor $\phi$, we adopt the GCN implementation from Yang et al. (2019). We use their default hyperparameters across all the datasets and baselines. Specifically, the GCN contains three convolution layers with hidden dimension 300. The predictor $f$ is a two-layer MLP with hidden dimenion 300 and ReLU activation. The model is trained with Adam optimizer for 30 epochs with batch size 50 and learning rate $\eta$ linearly annealed from $10^{-3}$ to $10^{-4}$. For RGM, we explore $\lambda \in \{0.01, 0.1\}$ for each dataset. For SRGM, we explore $\lambda_g \in \{0.1, 1\}$ for the classification datasets while $\lambda_g \in \{0.01, 0.1\}$ for the QM9 dataset as $\lambda_g = 1$ causes gradient explosion.

**Scaffold Classification** The scaffold classifier is trained by negative sampling since scaffolds are structured objects. Specifically, for each molecule $x_i$ in a minibatch $B$, the negative samples are the scaffolds $\{s_k\}$ of other molecules in the minibatch. The probability that $x_i$ is mapped to its correct scaffold $s_i$ is then defined as

$$p(s_i \mid x_i, B) = \frac{\exp\{g(\phi(x_i))^\top g(\phi(s_i))\}}{\sum_{k \in B} \exp\{g(\phi(x_i))^\top g(\phi(s_k))\}} \tag{26}$$

The scaffold classification loss is $-\sum_i \log p(s_i \mid x_i, B)$ for a minibatch $B$. We choose the classifier $g$ to be a two-layer MLP with hidden dimension 300 and ReLU activation.

### B.2 PROTEIN MODELING

**Data** The homology and stability dataset are downloaded from Rao et al. (2019). The size of each training environment, validation and test set are listed in Table 5.

**Model hyperparameters** For both tasks, our protein encoder is a pre-trained BERT (Rao et al., 2019). The predictor is a linear layer and the superfamily/topology classifier is a two-layer MLP whose hidden layer dimension is 768. The model is fine-tuned with an Adam optimizer with learning rate $10^{-4}$ and linear warm up schedule. The batch size is 16 and 20 for the homology and stability task. For RGM and SRGM, we explore $\lambda \in \{0.01, 0.1\}$ and $\lambda_g \in \{0.1, 1\}$ respectively.

Table 6: Mean absolute error (MAE) on the QM9 dataset under molecular size split. Models are trained on molecules with no more than 7 atoms and tested on molecules with 9 atoms. Due to space limit, we only show standard deviation for the top three methods in subscripts.

| | Categorical methods | | | | | | Structured methods | |
| Property | ERM | DANN | CDAN | IRM | MLDG | RGM | CrossGrad | SRGM |
|---|---|---|---|---|---|---|---|---|
| mu | 0.658 | 0.655 | 0.655 | 0.690 | **0.654** | $0.656_{(.004)}$ | $0.664_{(.001)}$ | $0.666_{(.005)}$ |
| alpha | 13.08 | 13.17 | 13.19 | 13.16 | 14.13 | $\mathbf{12.99}_{(.028)}$ | $12.79_{(.379)}$ | $\mathbf{11.54}_{(.777)}$ |
| HOMO | 0.008 | 0.008 | 0.008 | 0.009 | 0.008 | $0.008_{(.000)}$ | $0.008_{(.000)}$ | $0.009_{(.000)}$ |
| LUMO | 0.011 | 0.011 | 0.011 | 0.011 | 0.011 | $\mathbf{0.010}_{(.000)}$ | $0.011_{(.000)}$ | $0.013_{(.000)}$ |
| gap | 0.014 | 0.013 | 0.014 | 0.015 | 0.014 | $\mathbf{0.012}_{(.001)}$ | $0.014_{(.001)}$ | $0.016_{(.001)}$ |
| R2 | 352.8 | 355.7 | 357.3 | 368.6 | 381.2 | $\mathbf{328.4}_{(11.2)}$ | $351.7_{(11.0)}$ | $\mathbf{279.9}_{(29.6)}$ |
| ZPVE | 0.025 | 0.024 | 0.025 | 0.025 | 0.026 | $\mathbf{0.022}_{(.000)}$ | $0.024_{(.001)}$ | $\mathbf{0.019}_{(.001)}$ |
| Cv | 5.336 | 5.351 | 5.369 | 5.327 | 5.756 | $\mathbf{4.860}_{(.228)}$ | $5.235_{(.176)}$ | $\mathbf{3.909}_{(.420)}$ |
| U0 | 67.18 | 67.57 | 67.34 | 67.67 | 71.83 | $\mathbf{60.25}_{(2.62)}$ | $63.82_{(1.82)}$ | $\mathbf{51.32}_{(4.51)}$ |
| U | 66.67 | 67.00 | 67.24 | 68.55 | 71.60 | $\mathbf{58.74}_{(2.51)}$ | $64.30_{(1.47)}$ | $\mathbf{51.54}_{(5.09)}$ |
| H | 67.00 | 67.39 | 67.27 | 68.23 | 71.47 | $\mathbf{59.72}_{(2.23)}$ | $64.39_{(2.19)}$ | $\mathbf{50.17}_{(2.56)}$ |
| G | 65.92 | 65.95 | 66.02 | 68.16 | 70.70 | $\mathbf{59.40}_{(2.12)}$ | $64.63_{(1.12)}$ | $\mathbf{51.23}_{(6.13)}$ |

Table 7: SRGM performance under different molecular size split.

| Train | mu | alpha | homo | lumo | gap | R2 | zpve | Cv | U0 | U | H | G |
|---|---|---|---|---|---|---|---|---|---|---|---|---|
| 6 atoms | 0.92 | 21.5 | 0.010 | 0.016 | 0.018 | 521.6 | 0.035 | 7.73 | 100.1 | 99.4 | 102.0 | 100.1 |
| 7 atoms | 0.67 | 11.5 | 0.009 | 0.013 | 0.016 | 279.9 | 0.019 | 3.91 | 51.3 | 51.5 | 50.17 | 51.23 |
| 8 atoms | 0.69 | 4.00 | 0.007 | 0.009 | 0.011 | 119.0 | 0.009 | 1.59 | 20.2 | 20.1 | 19.7 | 20.5 |

### B.3 ADDITIONAL EXPERIMENTS

For the quantum chemistry dataset (QM9), prior work (Chen et al., 2019) has proposed to measure domain generalization via molecular size split. To show that our method also works well under this evaluation setup, we split the dataset based on the number of heavy atoms. The training set contains molecules with no more than 7 heavy atoms. The validation and test set consist of molecules with 8 and 9 heavy atoms respectively. This setup is much harder than random split as it requires models to extrapolate to new chemical space.

Our results on the QM9 dataset are shown in Table 6. Among the categorical methods, RGM outperforms all the baselines (except for property mu), with significant improvement on six properties (R2, Cv, U0, U, H, G) with 7-10% relative error reduction. SRGM outperforms all the baselines on eight properties (out of 12). While CrossGrad utilizes scaffold information, its performance is worse than RGM in general. Compared to RGM, SRGM shows significant error reduction (10-20%) on seven properties (alpha, R2, Cv, U0, U, H, G). This validates the advantage of exploiting structures of the environments (scaffolds).

We further conduct additional experiments to study the performance of RGM/SRGM with respect to the severity of domain shift. Fixing the test set to molecules with 9 atoms, we construct three progressively harder training sets: molecules with no more than 8, 7 and 6 atoms. We report the MAE ratio (averaged over 12 properties) between SRGM/RGM/CrossGrad and ERM. As shown in Figure 6, SRGM consistently outperforms CrossGrad and RGM across different setups.

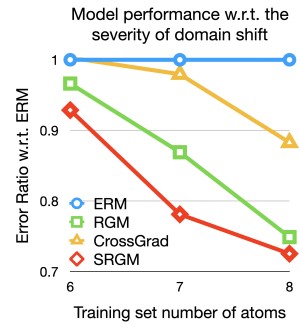

Figure 6: QM9 ablation study

