# OpenReview forum: "Enforcing Predictive Invariance across Structured Biomedical Domains"
_ICLR.cc/2021/Conference — Reject_

### Official Review · AnonReviewer3 · 2020-10-22
**Interesting idea, solid theoretical analysis.**

**Rating:** 6
**Confidence:** 4

**Review:**



Summary: The paper proposed a new  regret minimization (RGM) algorithm on structured environment. It is based on one existing SOTA work --- invariant risk minimization (IRM), which assumes the predictor is simultaneously optimal across various environments. To relax the restriction, the authors propose to introduce held-out environment to perform domain perturbation. The experiments are conducted on real-world biomedical applications including molecule property prediction and protein homology and stability prediction. The proposed method outperform baseline methods.



Strength

The paper proposed a novel Regret minimization (RGM) algorithm that extends IRM by relaxing the restrictive assumption on IRM, reformulate the problem as a joint optimization problem including regret minimization. The usage of held-out environment to encourage domain generalization looks very interesting.

The paper has solid theoretical analysis, and also demonstrate that the predictor obtained by RGM satisfies IRM constraints in the general case.  Empirical studies are convincing. Experiments on two real-world biomedical applications are of high value. The selected baselines are thorough and all from very recent paper. The performance gain looks significant, consistently outperforming baseline methods.


Weakness:

1. The author argues that their methods works in both standard and structured setting in Introduction Section, but only structured prediction is done.

2. MLDG is very relevant to the proposed method. It would be great if the author can discuss its difference with RGM in details, instead of mentioning it only in Related Work Section.

3. The presentation of Section 3 could be improved - notations could be better explained.

4. In theoretical analysis, can authors add the discussion about \phi_*? Now all the discussion are based on \phi \in \Phi_{IRM/RGM}.

5. I am not sure if the assumptions made in Section 3 are restrictive. For example, “new environments we may encounter at test time exhibit similar variability as the training environments”. Can you add some reference?

---

> ### Author Response · Authors · 2020-11-21
> **Thank you for your valuable feedback!**
>
> We have improved the presentation of section 3 with better explanation of notations.
>
> Q1: The author argues that their methods work in both standard and structured settings in the introduction section, but only structured prediction is done.
>
> In Table 1, we also report the performance of RGM under the standard setting. Note that all the categorical methods (DANN, CDAN, IRM, MLDG and RGM) do not utilize the scaffold information.  Among the categorical methods, RGM achieves the lowest MAE on 9 properties (mu, alpha, LUMO, gap, ZPVE, U0, U, H, G), with a clear improvement on U0, U, H and G. In Table 2, we also report the performance of RGM on the protein homology dataset. Its top-1 accuracy is 23.4%, outperforming all the other categorical baselines.
>
> Q2: MLDG is very relevant to the proposed method. It would be great if the author can discuss its difference with RGM in detail.
>
> The key difference between MLDG and RGM is the introduction of regret. MLDG requires the model to perform well on the meta-test domains after meta-update on the meta-train domains. This is similar to leave-one-out cross validation at the domain level. The regret term in RGM also requires the model to perform well on the meta-test domains, but it further requires it to be “optimal”. This introduces an additional term which forces the meta-test oracle predictor $f_e$ to perform no better than meta-train predictor $f_{-e}$. This also makes the RGM objective a mini-max game while MLDG is not.
>
> Q3: In theoretical analysis, can authors add the discussion about $\phi_*$? Now all the discussions are based on $\phi \in \Phi_{IRM/RGM}$.
>
> We suppose that $\phi_*$ means an optimal solution of RGM. If it achieves zero regret, then  $\phi_* \in \Phi_{RGM}$ by definition (Equation 8). It is difficult to give theoretical analysis when $\phi_*$ has non-zero regret and we leave this for our future work.
>
> Q4:I am not sure if the assumptions made in Section 3 are restrictive. Can you add reference for “new environments we may encounter at test time exhibit similar variability as the training environments”?
>
> We believe this assumption is important for domain generalization. By no free lunch theorem, we cannot expect the method to generalize to new domains arbitrarily different from the training domains. Our assumption essentially states that there exists some common factors in the source domains that will persist to new testing domains [1,2,3].
>
>
> References
>
> [1] Muandet, Krikamol, David Balduzzi, and Bernhard Schölkopf. "Domain generalization via invariant feature representation." International Conference on Machine Learning. 2013.
>
> [2] Ghifary, Muhammad, et al. "Domain generalization for object recognition with multi-task autoencoders." Proceedings of the IEEE international conference on computer vision. 2015.
>
> [3] Li, D.; Yang, Y.; Song, Y.-Z.; and Hospedales, T. M.. Deeper, broader and artier domain generalization. In ICCV. 2017

---

### Official Review · AnonReviewer2 · 2020-10-27
**Official Blind Review #2**

**Rating:** 4
**Confidence:** 3

**Review:**

##########################################################################

Summary:
The authors proposed regret minimization (RGM) and structured RGM (SRGM) algorithms for generalization across biomedical domains. They quantified generalization to held-out environments by using two auxiliary predictors trained with and without each environment. Furthermore, for structured environments, they simultaneously trained an auxiliary scaffold classifier and used it to generate perturbations highlighting the environmental variation. The authors evaluated the proposed method on several tasks from molecular property prediction, protein homology, and stability prediction. They showed significant performance improvement over current state-of-the-art algorithms.

##########################################################################

Major comments:
While the paper has its own merits, unfortunately, it has several issues that need to be addressed.
-	My main concern is that I am not sure the experiment settings are truly realistic. I agree with the authors’ motivation that generalizing beyond training domains and environments holds great importance. However, in my view, the current experiment setting does not quite represent real-world applications. For molecular property prediction, the authors used the number of atoms or molecular weights to determine the scaffold of data. However, in real-world settings, I do not think it is common to have training data with small molecules and test data with larger molecules. Wouldn’t it be more realistic to determine scaffolds based on sharing a subgraph of a molecular graph as described in Figure 1?
-	In the case of stability prediction, the train/validation/test splits are not based on the topology. As stated by the authors, the test split contains Hamming distance-1 neighbors of around those from the training set. I am not sure whether the dataset is appropriate to be used to evaluate the proposed method for generalization beyond training domains.
-	The key component of SRGM compared to RGM would be the adoption of a scaffold classifier. Then, I think the performance of SRGM should be inevitably dependent on the performance of the scaffold classifier. Can you provide the classification performance of the scaffold classifier? Assuming that the top-1 accuracy for superfamily classification is similar to that in TAPE (which is only about 40%), can you explain how can SRGM provide performance improvement with such an inaccurate classifier? Furthermore, if most scaffolds contain only few examples, wouldn’t the training of the scaffold classifier be unstable, thus, lower the performance of SRGM as well?
-	I think the ablation studies with SRGM-detach showed some related results, can you show how random perturbations instead of using the scaffold classifier would affect the performance of the proposed method?

##########################################################################
Minor comments:
-	In protein modeling experiments, how did the authors generate sequence-length invariant protein embeddings from the BERT representations? Did you adopt an additional layer to compute attention-weighted mean protein embeddings as used in Rao et al.?
-	I understand that the authors omitted detailed explanations of each task and data due to space limitations. However, to be self-contained, I would like to recommend to include more information in the appendix. For example, the number of labels for the HOMO task is not stated in the paper.
-	Can you provide a more concise definition of standard and structured environments?
-	It is obvious that SRGM stands for structured RGM, but anyhow it appeared in the manuscript without proper abbreviation explanation.

##########################################################################

---

> ### Author Response · Authors · 2020-11-21
> **Thank you for your insightful comments!**
>
> We have improved the presentation of section 3 with better explanation of notations and abbreviations.
>
> Q1: Are the experiment settings truly realistic? In real-world settings, I do not think it is common to have training data with small molecules and test data with larger molecules.
>
> For the quantum chemistry dataset (QM9), prior work (Chen et al., 2019, Tsubaki et al., 2020) has proposed to measure model generalization by splitting the dataset via number of atoms. Molecular weight split has also been adopted by Feinberg et al. (2019) for medicinal chemistry. We initially followed their splitting methods in our experiments. Nevertheless, we agree that these two splitting methods are not perfect ways to measure domain generalization as the test molecules can be much bigger. The major challenge is that standard scaffold split degenerates to random split when most scaffold clusters contain only one molecule. Therefore, scaffold split fails to measure domain generalization.
>
> Given these observations, we propose a variant of scaffold split called scaffold complexity split. We define the complexity of a scaffold as the number of cycles in the scaffold graph. Specifically, we put a molecule in the test set if its scaffold has a high complexity and the training set if its scaffold complexity is low (the scaffold is defined as the Murcko scaffold illustrated in Figure 1).
> This forces the test scaffolds to be structurally different from the training scaffolds. As shown in Figure 5, the molecular weight distribution of the training and test molecules are similar. We hope this serves as a more realistic evaluation.
>
> Q2: In the case of stability prediction, the train/validation/test splits are not based on the topology. Not sure whether this dataset is appropriate to evaluate the proposed method.
>
> We have removed this dataset from the paper so that all evaluation setups are coherent.
>
> Q3: What is the classification performance of the scaffold classifier? Can you explain how SRGM provides performance improvement when superfamily classification accuracy is low? If most scaffolds contain only a few examples, wouldn’t the training of the scaffold classifier be unstable and hurt the performance of SRGM?
>
> The scaffold classification performance is listed in Table 2 (right). On the molecule datasets, the scaffold classification accuracy is over 90%. On the protein homology dataset, the superfamily classification accuracy is 33.5% (top-1) and 51.0% (top-10). However, SRGM can still give performance improvement because during training, the gradient perturbation is computed based on the ground truth superfamily label. This teacher forcing step helps SRGM to be robust to superfamily variation despite the inaccurate superfamily classifier.
>
> Q4: Can you show how random perturbations would affect the performance of the proposed method?
>
> In Table 3, we report the performance of SRGM under random perturbation on the QM9 dataset. Random perturbation performs significantly worse for most of the properties.
>
> Q5:  In protein modeling experiments, how did the authors generate sequence-length invariant protein embeddings?
>
> We simply take the mean of all the protein sequence vectors. Our ERM baseline with average-pooling achieves the same performance as TAPE on the homology dataset.
>
> Q6: Detailed explanations of each task.
>
> We have added the explanations of each task in section 4.1 as well as the number of homology labels in section 4.2.
>
> Q7: Can you provide a more concise definition of standard and structured environments?
>
> An environment $E_k$ is called categorical if $k$ is an integer. $E_k$ is structured if $k$ is a structured object such as molecular scaffold.
>
> Reference
>
> [1] Chen, Guangyong, et al. "Alchemy: A quantum chemistry dataset for benchmarking ai models." arXiv preprint arXiv:1906.09427 (2019).
>
> [2] Tsubaki, Masashi, and Teruyasu Mizoguchi. "Quantum Deep Field: Data-Driven Wave Function, Electron Density Generation, and Atomization Energy Prediction and Extrapolation with Machine Learning." Physical Review Letters 125.20 (2020): 206401.
>
> [3] Evan N Feinberg,  Robert Sheridan,  Elizabeth Joshi,  Vijay S Pande,  and Alan C Cheng.   Stepchange improvement in admet prediction with potentialnet deep featurization. arXiv preprintarXiv:1903.11789, 2019.

---

### Official Review · AnonReviewer1 · 2020-10-27
**Interesting work but the authors should proofread the manuscript further**

**Rating:** 5
**Confidence:** 4

**Review:**

In bioinformatics and biochemistry, we may need to generalize the model beyond the training distribution. The authors propose a new method, the regret minimization (RGM) algorithm, to handle such a problem. The method is inspired by invariant risk minimization (IRM). It regulates the empirical loss with the regret regularizer. Such a regularizer encourages the model to generalize to a new unseen environment. The authors further extend the RGM to the structured inputs, leading to structured RGM (SRGM), which can be trained using gradient perturbation. Overall, the manuscript is interesting, which may be suitable for the general audience of ICLR.

Pro:
1. Interesting problem and a new self-contained solution.
2. Promising performance across different tasks from different domains.
3. The manuscript is easy to follow although I believe it could be improved further.

Concerns:
1. The authors should further proofread the manuscript. I guess on page 5, the authors wanted to say that "the model would not see any change from one example to another as scaffold variation", right? Some symbols in the paper are not defined as well. For example, in Figure 2, I did not find the definition for $B_i$, $B_j$, and $B_e$. I am willing to increase the score if the manuscript is refined further.
2. On page 6, my feeling is that the model is not very difficult to train. Are there any tricks to train such a model? Is it similar to adversarial training?
3. In Table 2, why we have both RGM and SRGM? What's the difference between the two methods within this table? If I understand correctly, all the inputs are structured, right?
4. In Figure 4, wow much would SRGM decrease for the different number of atoms? I mean the absolute value, not the relative value to ERM.
5. For the HOMO, the authors include data from different protein families. How can the method generalize across different families?
6. Regarding the feature extractor $\phi$, how would the model complexity affect the performance of the proposed method? Any suggestions on the selection of the parametric family?

---

> ### Author Response · Authors · 2020-11-21
> **Thank you for your insightful comments!**
>
> Q1: The authors should further proofread the manuscript. Some notations are undefined.
>
> We apologize for the confusion. On page 5, what we wanted to say is that when an environment has only one molecule, the model cannot decide which subgraph of that molecule is the right scaffold. Therefore, creating single-example environments is not helpful for domain generalization. We have updated this paragraph in the paper.
> In Figure 2, $B_i, B_j, B_e$ are three mini-batches sampled from environments $E_i, E_j, E_e$. To make the flow better, we have split Figure 2 into two figures. RGM backward pass is now illustrated in Figure 4 in page 6 so that it is coupled with the optimization section (sec 3.3). We have also refined the method section and resolved all the undefined notations.
>
> Q2: RGM seems difficult to train. Are there any tricks to train this model? Is it similar to adversarial training?
>
> Training RGM is not difficult. The only “trick” we adopted is the gradient reversal layer between the oracle predictor $f_e$ and the feature extractor $\phi$. This is important because they optimize the loss $\mathcal{L}^e(f_e \circ \phi)$ in different directions. This allows us to update all the players in a single forward-backward pass, just like domain adversarial training (Ganin et al., 2016).
>
> Q3: In Table 2, why do we have both RGM and SRGM? What's the difference between the two methods within this table?
>
> Each environment $E$ is composed of a set of molecules and associated with a structure label (i.e., scaffold). RGM does not utilize the structure of the environment and simply treats an environment as a set of molecules. SRGM uses the scaffold graph to make perturbation in the representation $\phi$. In other words, SRGM utilizes extra structural information and has the advantage over RGM. We thus provided the result for both RGM and SRGM.
>
> Q4: In Figure 4, how much would SRGM decrease for the different number of atoms?
>
> The result is reported in Table 7 in the appendix. The absolute MAE difference varies for different properties. For instance, U0 goes from 51.3 to 23.9 when the number of atoms in the training set increases from 7 to 8.
>
> Q5: For the homology task, the authors include data from different protein families. How can the method generalize across different families?
>
> SRGM learns to generalize across different protein superfamilies through perturbation from the superfamily classifier. Each protein is assigned to a superfamily and the superfamily classifier learns to predict its superfamily from its protein sequence.
>
> Q6: Regarding the feature extractor, how would the model complexity of $\phi$ affect the performance of the proposed method?
>
> To study how the model complexity of $\phi$ affects the performance of SRGM, we train SRGM under different numbers of graph convolutional layers on the QM9 dataset. As shown in Table 4, SRGM performs the best when there are three graph convolutional layers. Similar to standard ERM, SRGM underfits the data when the model is too simple (layer=2) and overfits when the model is too complex (layer=4). For the protein homology dataset, the feature extractor is a pre-trained BERT with fixed architecture and thus we cannot run ablation studies for it.

---

### Official Review · AnonReviewer4 · 2020-10-30
**A new regret minimization algorithm applied for structured environments**

**Rating:** 5
**Confidence:** 3

**Review:**

Summary:
This work is built on top of IRM, aiming to solve an issue in the biomedical domain where there are very few instances in a given environment. The authors first proposed a new scheme, RGM, to apply tighter constraints on the generalization. And then they described a gradient perturbation scheme in the representation space to corrupt the environment information carried in the representation. The authors claimed improved performance over existing methods on several empirical experiments using biomedical benchmarks.

Strength:
1) RGM and a perturbation scheme for structured environments with few instances. Some theoretical analyses showing RGM imposes stricter constraints than IRM.
2) Pretty comprehensive evaluation on several benchmark datasets to show the model performance/generalization
3) The flow of the paper is clear, not hard to follow

Weakness
1) The contribution and significance are not clear to me. Both RGM and perturbation (esp. compared to CrossGrad) do not seem very novel or significant. Maybe the authors could elaborate on this more.
2) While the authors described the hyperparameter search in the appendix, it would be nice to know more about the auxiliary models and losses. For example, how well does the scaffold classifier perform in the experiments as well as in the ablation studies (i.e. fig 5)?
3) The data split based on (heavy) atom numbers, scaffold molecule weights and #instances in a protein superfamily look a bit weird. Does this truly probe the model generalization? In other words, does a big molecule weight difference necessarily correspond to two very distinct scaffolds?
4) Figures are not in the correct order (fig 2). Understandable this is likely to accommodate the page limits but is causing some confusion.

---

> ### Author Response · Authors · 2020-11-21
> **Thank you for your insightful comments!**
>
> Q1: Both RGM and perturbation (esp. compared to CrossGrad) do not seem very novel.
>
> The closest method to RGM is IRM. There are two key differences between RGM and IRM.
> We replace the IRM simultaneous optimality constraint by regret. This is important for over-parameterized models which can easily achieve zero training error. If an ERM optimal solution achieves zero training error, it will be simultaneously optimal in all training environments and thus also an IRM optimal solution. RGM avoids this problem because $f_{-e}$ is not trained on held-out domains and therefore does not achieve zero error.
> IRM simplifies the simultaneous optimality constraint to a gradient penalty term, which forces the predictor to be a constant function. In contrast, RGM is formulated as a multiplayer game, which allows the predictor to be non-linear neural networks.
>
> The difference between SRGM and CrossGrad lies in the way perturbation is used. In CrossGrad, the model is trained to minimize its error on the perturbed examples. In SRGM, the model is trained to minimize its predictive regret on the perturbed examples (i.e., how well it can generalize after unseen perturbation). This forces the model to be optimal in the environment of perturbed examples, which is a stronger constraint than CrossGrad.
>
> Q2: Performance of the scaffold classifier
>
> The performance of the scaffold classifier is reported in Table 2. We report the scaffold classification accuracy for SRGM as well as its variants in the ablation studies. Overall, the scaffold classification accuracy is over 90% in the molecule datasets.
>
> Q3: The data split based on atom numbers, scaffold molecule weight and #instances in a protein superfamily looks weird. Does this truly probe model generation? Does a big molecule weight difference necessarily correspond to two very distinct scaffolds?
>
> In response to other reviewer’s feedback, we have changed the molecular property prediction experiments in order to provide a more realistic evaluation of domain generalization. Specifically, we propose to split the dataset via scaffold complexity split. We define the complexity of a scaffold as the number of cycles in the scaffold graph. We put a molecule in the test set if its scaffold has a high complexity and the training set if its scaffold complexity is low. As a result, the training and test scaffolds are different (see Figure 5). Under this new evaluation setup, SRGM still outperforms other methods across all datasets.
>
> Q4: Figures not in the correct order (fig 2).
>
> We have split Figure 2 into two figures. RGM backward pass is now illustrated in Figure 4 in page 6 so that it is coupled with the optimization section (sec 3.3). Now the figures are in order.

---

### Author Response · Authors · 2020-11-21
**Paper updated**

We want to thank all the reviewers for your insightful comments. In summary, we have updated the manuscript with the following changes:
1. We have refined section 3 and resolved all the undefined notations and unclear statements. In particular, we split Figure 2 into two figures. RGM backward pass is now illustrated in Figure 4 in page 6 so that it is coupled with the optimization section (sec 3.3).
2. We have changed the molecular property prediction experiments in order to provide a more realistic evaluation. For the QM9 experiment, originally we followed Chen et al., 2019 and split the data based on the number of heavy atoms, but this setup does not probe generalization in terms of scaffold changes. Instead, we propose to split the data via scaffold complexity split (detailed in section 4.1). As a result, the training and test scaffolds are structurally different while the molecular weight distribution of the training and test molecules are similar (see Figure 5). The original experiments are moved to the appendix for bookkeeping purposes.
3. We have removed the protein stability dataset to avoid confusion. Reviewer 2 has pointed out that the train/validation/test splits are not based on the protein topology.
4. We also report the scaffold / protein superfamily classification accuracy in Table 2.

Reference

Chen, Guangyong, et al. "Alchemy: A quantum chemistry dataset for benchmarking ai models." arXiv preprint arXiv:1906.09427 (2019).

---

### Decision · Program_Chairs · 2021-01-07
**Final Decision**

**Decision:**

Reject

**Comment:**

The paper has some interesting points in extending IRM to regret minimization, and extending to structured environments.  I can see the writing has been improved in the revision.  The main criticism arises from the experiment, which can be improved in several aspects.   The reviews have been quite detailed and helpful.